# A computational predictor of the anaerobic mechanical power outputs from a clinical exercise stress test

Efrat Leopold, Tamir Tuller[☯], Mickey Scheinowitz[iD]*[☯]

Department of Biomedical Engineering, Tel-Aviv University, Tel Aviv, Israel

☯ These authors contributed equally to this work.
* mickeys@tauex.tau.ac.il

**Data Availability Statement:** All relevant data are within the manuscript and its Supporting Information files.

**Funding:** This study was partially supported by the Elizabeth and Nicholas Slezak Super Center for

## Abstract

We previously were able to predict the anaerobic mechanical power outputs using features taken from a maximal incremental cardiopulmonary exercise stress test (CPET). Since a standard aerobic exercise stress test (with electrocardiogram and blood pressure measurements) has no gas exchange measurement and is more popular than CPET, our goal, in the current paper, was to investigate whether features taken from a clinical exercise stress test (GXT), either submaximal or maximal, can predict the anaerobic mechanical power outputs to the same level as we found with CPET variables. We have used data taken from young healthy subjects undergoing CPET aerobic test and the Wingate anaerobic test, and developed a computational predictive algorithm, based on greedy heuristic multiple linear regression, which enabled the prediction of the anaerobic mechanical power outputs from a corresponding GXT measures (exercise test time, treadmill speed and slope). We found that for submaximal GXT of 85% age predicted HRmax, a combination of 3 and 4 variables produced a correlation of r = 0.93 and r = 0.92 with % error equal to 15 ± 3 and 16 ± 3 on the validation set between real and predicted values of the peak and mean anaerobic mechanical power outputs (p < 0.001), respectively. For maximal GXT (100% of age predicted HRmax), a combination of 4 and 2 variables produced a correlation of r = 0.92 and r = 0.94 with % error equal to 12 ± 2 and 14 ± 3 on the validation set between real and predicted values of the peak and mean anaerobic mechanical power outputs (p < 0.001), respectively. The newly developed model allows to accurately predict the anaerobic mechanical power outputs from a standard, submaximal and maximal GXT. Nevertheless, in the current study the subjects were healthy, normal individuals and therefore the assessment of additional subjects is desirable for the development of a test applicable to other populations.

## Introduction

Exercise training and intervention programs are increasingly being proposed in daily practice with the aim to increase aerobic / cardiorespiratory fitness level (CRF) and maintaining general health [1–4]. A standard graded exercise test (GXT) include incremental increase [5]

Cardiac Research and Medical Engineering, Tel Aviv University (MS).

Competing interests: The authors have declared that no competing interests exist.

which terminate at submaximal or maximal intensities based on age-adjusted maximal HR [6]. At such intensities part of the anaerobic metabolic reservoir, of ATP-PC and glycolysis, are being used. Numerous studies [7–9] have previously shown the relationship between metabolic contents / reservoir and anaerobic mechanical power outputs using various suggested anaerobic stress tests, including the Wingate anaerobic test (WAnT). Since measuring phosphagen and glycogen contents is complicated and required muscles samples [10, 11], these metabolic systems are usually reflected via power output measurement using the WAnT. Therefore, since knowing the anaerobic components is important, performing an anaerobic exercise test is required. We recently have shown a high predictive model of the peak and mean anaerobic mechanical power outputs using gas exchange indices during maximal incremental cardiopulmonary exercise testing (CPET) [12]. Yet, since GXT is more common and generally being used in the clinic (to assess hypertension, coronary disease, etc.), predicting of the anaerobic components, by mean of peak and mean anaerobic mechanical power outputs, from such a test would result in performing a single test that will generate data on both aerobic and anaerobic capacities. Prediction of the anaerobic mechanical power output will allow an easy way to understand the contribution of the anaerobic components using only one single exercise stress test. This information is of special interest for athletes [13–15], exercise physiologist and coaches who include anaerobic exercise in their training routine.

The integration of prediction model with physiological variables have been widely used. For example, George et al [16], used regression analysis for the prediction of maximal $VO_2$, based on submaximal treadmill exercise and non-exercise data. They found that the regression model can accurately predict maximal $VO_2$ using submaximal treadmill test, in healthy men and women from both exercise and non-exercise data. Luttikholt et al [17], used the critical power profile to develop a model enabling the prediction of anaerobic peak power (PP) output from the results of different GXT protocols and found no differences between the actual and the predicted PP output values. In this paper we aim to predict the anaerobic mechanical power outputs from a GXT, which has no gas exchange measurements. Our assumption was that during GXT, the subject reaches maximal or submaximal intensity, and at this point, anaerobic metabolism becomes dominate and allows the generation of additional mechanical power output [18]. We therefore hypothesized that during a standard GXT, there is information of anaerobic components, which can be directly translated into anaerobic mechanical power outputs using newly developed computational methods.

## Materials and methods

### Experimental design

We previously developed predictive model, which includes inputs of CPET features and outputs of anaerobic mechanical power [12]. More specifically, 51 features, from gas exchange analysis driven from a maximal incremental CPET were measured and calculated and used as input on the predictive algorithm. Eventually, the PP and mean power (MP) outputs form the Wingate anaerobic test (WAnT) were predicted, at which the predicted correlation were r = 0.94 and r = 0.9, respectively [12]. In the current paper, the same data that was collected from subjects performing both CPET and WAnT was used [12], yet the CPET data was adjusted/re- calculated in a way that represents a standard EST (which do not include gas exchange data). More specifically, the subject's **age** was used to calculated heart rate (HR) max (100%) based on Tanaka H et al. [19] formula, which equal to 208–0.7 x age. We then took the calculated HRmax value and superimposed that value with the treadmill's speed and slope obtained from the CPET. Those data served us to calculate **$VO_2max$** based on ACSM'S

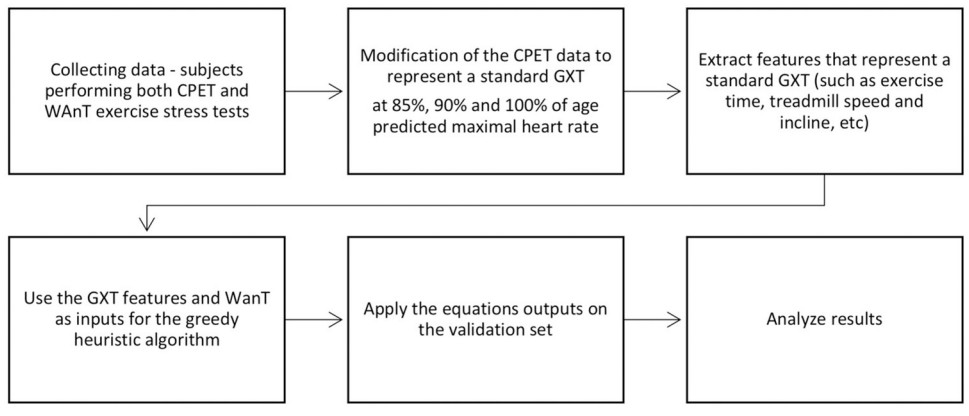

**Fig 1. The flow of the research.**

metabolic calculations handbook:

$$(1) \quad \text{VO2max} = (0.2 \cdot \text{Speed}) + (0.9 \cdot \text{Speed} \cdot \text{Grade}) + 3.5.$$

Since "clinical ergometry" usually ends when the subject reaches 85% to 90% of his/her HRmax [20], we calculated the 90% and the 85% HRmax values to reflect submaximal HR values. Those calculated values were used as features on the predictive algorithm to study the ability of standard aerobic features to predict the anaerobic mechanical power outputs. The study flow is presented in Fig 1.

## Data collection

Data was collected from normal, healthy male (n = 53) and female (n = 40) athletes and non-athletes, as described in previous paper [12], who performed both the Wingate test (WAnT), which is considered as the most popular anaerobic test [21, 22], and a maximal incremental CPET at the Zinman College for Physical Education and at the Department of Research and Sports Medicine, Wingate Institute [12]. All methods, including the experimental protocol, were performed in accordance with the relevant guidelines and regulations. Each subject signed an informed consent form to participate in the study. Briefly, each participant performed the WAnT and a CPET. The WAnT included a 3-minute warm up on a cycle ergometer at moderate intensity and then the subjects performed an "all-out" 30 seconds test against constant resistance that was determine based on the participant weight times 0.075. Each subject performed the WAnT and CPET (2 weeks apart) one time, and therefore the learning effect was not relevant in the current protocol. The outputs of the WAnT are: (1) PP output which was defined as the maximal mechanical power output achieved during the first few seconds of the test (1–5 sec) (2) MP output which was defined as the average power throughout the 30 sec WAnT. And (3) fatigue index (%) which was calculated as the percent difference between the PP and the minimal mechanical power output reached during the test [21, 22].

A running maximal incremental CPET was performed on electric treadmill. The subjects began with a short warm-up period between 2–3 minutes, following by gradually increasing the speed by 1 [kilometer/hour] every minute, up until the respiratory quotient ($VCO_2/VO_2$) reached 1, and then the slope was increased by 2%. During the entire test, both the ventilatory and metabolic variables (such as $VCO_2$ and VE) were recorded for each breath, using the Quark CPET (COSMED, Rome, Italy) until the subjects reached maximal effort [12].

## Adapting features from CPET that represents GXT

We have used data taken from maximal incremental CPET and adjusted the data to create a new data-set which represent a standard, submaximal and maximal (age predicted [19]) GXT. In order to adjust the data. the following steps were implemented: (1) calculating age-predicted maximal heart rate (HR) [19]. (2) Calculating the HR at 100%, 90% and 85% of the maximal, age-predicted HR which represent maximal and sub-maximal efforts which are common end-points of a GXT. Then, (3), calculating features which include the calculated HRs (100%, 90% and 85%), the total exercise time of the GXT, and the treadmill slope and speed at the different exercise levels of the calculated HR. Based on this, we have extracted 25 new features representing standard GXT as detailed in S1 Table. The features were: maximal HR, $VO_2$ predicted at 100% of max HR (mL/min), $VO_2$ predicted at 100% of max HR (mL/min/Kg), Slope * speed at 100% of max HR, Max intensity at 100% of max HR, Exercise time at 100% of max HR, $VO_2$ at AT (mL/min) at 100% of max HR, $VO_2$ at AT (mL/min/kg) at 90% of max HR. Each feature was calculated at the 3 different exercise levels of the calculated HR.

## Algorithm to predict the anaerobic mechanical power outputs

We have used a previously developed greedy heuristic algorithm to study the ability of GXT features to predict the anaerobic mechanical power outputs [12]. In nutshell, all features were randomly divided into three groups which included the validation (30% of the data), calibration (40% of the data) and test subsets (30% of the data). The algorithm output is a linear regression mathematical equation, which represent the 'best' features that shows the minimum distance between the predicted to the observed values of the anaerobic mechanical power output. The predicted $VO_2$ max values (which was calculates based on the ACSM equation) [23] were used to ensure that even values representing all exercise levels are within each group. The steps of the algorithm included the following: Using the train subset, at each iteration, the $i$-th (i = numbers of features) feature from the aerobic matrix was selected to performed multiple linear regressions and then the coefficients obtained was used to calculate the predicted anaerobic mechanical power output parameters. The distance between the predicted values obtained from the updated predictor to the observed values of the anaerobic parameter (using the least squares method) was saved. The minimum distance between the observed to the predicted values was the selected feature. Then, the selected feature and the chosen coefficients were used on the test subsets to calculate the predicted anaerobic parameter, following by the calculation of Spearman correlation coefficient acquired from the predicted with the real value of the anaerobic parameter. The termination condition was the percent change of the adjusted $R^2$, which was either increased or larger than 0.5%. The algorithm output was then applied to the validation set and the spearman correlation coefficients of the predicted anaerobic mechanical power outputs parameters were calculated compared with the real values. We have used the algorithm on the different exercise levels: 100%, 90% and 85% of maximal age-predicted HR in order to simulate the conditions used under clinical routine tests.

The model was applied to 100 randomly selected validation sets in order to determine the mean ± SD of the percent (%) error. The % error was calculated using the following equation:

$$\%error = \frac{|Yreal - Ypredicted|}{Yreal} * 100 \tag{1}$$

$Y_{real}$ refers to the real value of the predicted feature–either PP or MP.

$Y_{prediced}$ refers to the predicted value that is calculated–either PP or MP.

**Table 1. Physical characteristics of the participants.**

| Variable | Female (N = 40) (Mean± SD) | Male (N = 53) (Mean± SD) |
|---|---|---|
| Age (y) | 26 ± 4 | 29 ± 6 |
| Body Height (cm) | 163.8 ± 6.4 | 176.6 ± 6.8 |
| Body Mass (kg) | 60.9 ± 9.8 | 75.7 ± 10 |
| Body Mass Index (Kg/m$^2$) | 22.7 ± 3.2 | 24.3 ± 2.7 |
| Calculated age-predicted HR max | 189.8 ± 3 | 187.9 ± 4 |

## Results

Demographic characteristic of the subjects is presented in Table 1.

The peak and mean mechanical power outputs prediction equations derived from the calibration sets, for each exercise levels, are presented in Tables 2 and 3, respectively. Overall, features that represented 85% of maximal age-predicted HR produced the highest predictions for PP of r = 0.93 (p < 0.001) and 15% error and features that represent 100% of maximal age-predicted HR produced the highest predictions for MP of r = 0.94 (p < 0.001) and 14% error. Furthermore, features representing 90% and 85% of maximal age-predicted HR produced a lower prediction for PP and MP of r = 0.90 and r = 0.92 (p < 0.001) and % error of 16 and 15, respectively. For PP at 100% age-predicted HR, the equation included four features and produced a spearman correlation coefficient prediction of r = 0.92 with % error of 12 between the predicted and real PP values of the validation set, as illustrates in the dot plot in Fig 2C. For MP predictions at 100% of age-predicted HR, two features were chosen, and the combination of these features produced a spearman correlation coefficient of r = 0.94 with % error of 14 between the predicted and real MP values of the validation set, illustrates in the dot plot and a bar diagram in Fig 3C. Yet, the differences between the different levels of age-predicted HR (85%-100%) of the predictions for PP and MP from the lowest to highest prediction correlation were ~3% and ~2%, respectively (Figs 2A-2C and 3A-3C).

## Discussion

The purpose of this paper was to investigate whether the prediction of the anaerobic mechanical power outputs using the limited features obtained in a standard GXT is feasible. The main observation of our current study is the ability to predict peak and mean anaerobic mechanical power outputs using a standard GXT. Due to the high prediction model and outcomes, gas exchange and VO$_2$ measurement is probably not necessary. Our current study suggests that scientists, physicians and coaches can use aerobic features (as shown in S1 Table) from a standard aerobic GXT with high accuracy to predict the peak and mean anaerobic mechanical power outputs. When comparing the results from the current study with the ones obtained from CPET gas exchange variables, we can see that the prediction for PP from features taken from CPET produced a spearman correlation coefficient of r = 0.95 [12] which is only 2%

**Table 2. Multiple linear regression equations together with the prediction correlation, P-value and RMSE of the predicted equations for the validation group, for peak power (w) at the different exercise levels equal to 85%, 90%, and 100%of maximal age-prediction HR.**

| Category | Multiply regression equation | | | R$^2$ and RMSE of validation set | | |
|---|---|---|---|---|---|---|
| | 85% | 90% | 100% | 85% | 90% | 100% |
| Peak Power (w) | 644 + Predicted VO$_2$ (ml/min) * 225—Predicted VO2 (ml/min/kg) * 87—slope * speed * time * 26 | 642 + Predicted VO$_2$ (ml/min) * 193—VO$_2$ at AT(ml/min/kg) * 99—slope * speed * time * 25 + VO$_2$ at AT (ml/min) * 34 | 645 + Predicted VO$_2$ * 225—Predicted VO2 (ml/min/kg) * 100 + slope * speed *67—slope * speed * time *32 | 0.93 (P = 5.53*e$^{-07}$), RMSE = 38.9 | 0.9 (P = 1.11*e$^{-06}$), RMSE = 100 | 0.92 (P = 6.87*e$^{-07}$), RMSE = 90 |

**Table 3. Multiple linear regression equations together with the prediction correlation, P-value and RMSE, of the predicted equations for the validation group, for mean power (w) at the different exercise levels equal to 85%, 90%, and 100%of maximal age-prediction HR.**

| Category | Multiply regression equation | | | $R^2$ and RMSE of validation set | | |
|---|---|---|---|---|---|---|
| | 85% | 90% | 100% | 85% | 90% | 100% |
| **Mean Power (w)** | 475 + Predicted $VO_2$ * 154 -$VO_2$ at AT (ml/min/kg) * 93 + time * 27 + $VO_2$ at AT (ml/min) * 6 | 478 + Predicted $VO_2$ (ml/min) * 167 –$VO_2$ at AT (ml/min/kg)* 101 + time * 33 | 478 + Predicted $VO_2$ (ml/min) * 156—Predicted $VO_2$ (ml/min/kg) * 39 | 0.92 (P = 7.64*e$^{-07}$), RMSE = 79 | 0.93 (P = 6.02*e$^{-07}$), RMSE = 78 | 0.94 (P = 9.43*e$^{-07}$), RMSE = 66 |

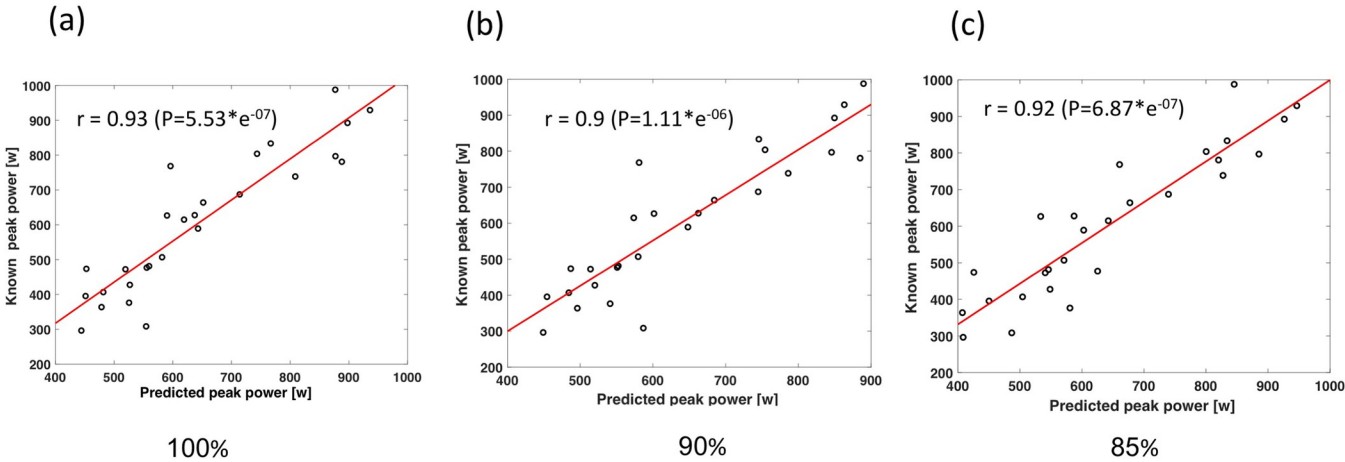

**Fig 2.** Line plot between the known versus predicted values of the validation set at different exercise levels for peak power at the different exercise levels equal to 85% (a), 90% (b) and 100% (c) of maximal age-predicted HR, for both male and female subjects.

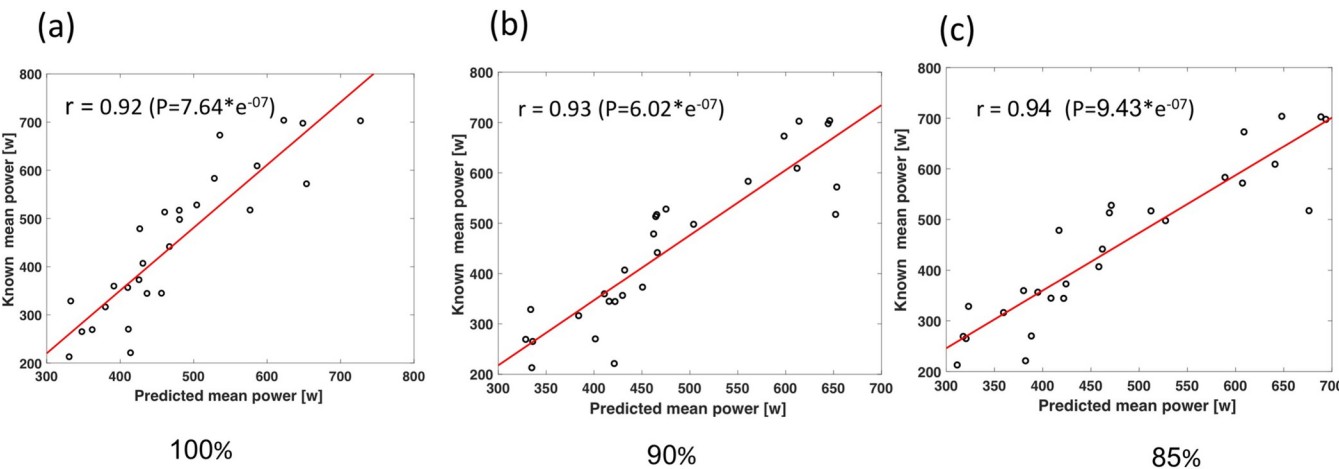

**Fig 3.** Line plot between the known versus predicted values of the validation set at different exercise levels for mean power at the different exercise levels equal to 85% (a), 90% (b) and 100% (c) of maximum, based on age-prediction HR max, for both male and female subjects.

higher than the PP predictions using features taken from standard GXT at 85% of age-prediction HR max and 3% higher than the PP predictions using features taken from standard GXT at 100% of age-prediction HR max. Furthermore, prediction of MP using features taken from CPET produced a spearman correlation coefficient of r = 0.91 which is lower than the MP predictions using features taken from standard GXT (at 100% of age-prediction HR max). Furthermore, the percent error of PP using features taken from CPET was 19, which is higher by

4% than features taken from standard GXT (at 85% of age-prediction HR max). The percent error of MP using features taken from CPET was 16, which is higher by 2% than features taken from standard GXT (at 100% of age-prediction HR max). Thus, prediction of the mechanical power outputs, PP and MP, is feasible using features from both maximal incremental CPET and standard GXT.

Looking into the rationale behind the chosen features in the predicted equations, each feature contributes at certain level to the prediction of the anaerobic mechanical power output. As we can see the $VO_2$max was chosen several times for the predicted equations. $VO_2$max was calculated using the American College of Sports Medicine (ACSM) metabolic equations [23], which aim to provide an accurate prediction of $VO_2$ max, and is based on maximal speed and slope at specific treadmill intensity. The $VO_2$ max is usually being used to describe exercise capacity in healthy individuals [24]. Following, since there is a portion of usage of aerobic metabolites during the WAnT [25], the $VO_2$ max feature might represent the aerobic contribution during the WAnT [26]. Another example is the VAT features which defines as the point at which changes in gas exchange in the lungs occur during increased exercise intensity (by means of blood lactate accumulation) [25]. Trained athletes accumulate less lactate during a fixed submaximal workload [27], hence this feature change depending on the individual fitness level. The presence of lactate in blood during exercise is the result of an increased glycogenolysis which is an indication of increase in glycogen catabolism. Therefore, the VAT feature can give an indication of lactate accumulation following glycogen metabolism, which is used during the WAnT [28].

The ability to predict the anaerobic mechanical power output will allow an easy way to understand the contribution of the anaerobic mechanical power outputs using a single exercise stress test. The information regarding the anaerobic components will provide a better understating of the subjects' physiological capabilities which is especially important for athletes, helping specify the aerobic and anaerobic contribution for exercise training recommendations. Furthermore, assessments of the anaerobic performance can help athletes, whether team players or individual competitors by providing the coach with valuable information about the athletes' fitness status as well as allowing them to monitor improvement through training.

One limitation of this study is the study population, which included healthy males and females within the age range of 20–40 years, yet exclude other populations such as athletes from various sports including aerobic and anaerobic phenotypes. Therefore, the data collection is not representative enough in order to produce automatic methods that can be implemented in commercial software, hence, more data is needed in order to produce a more generic model. Additional limitation of this study originated in the nature of the model: we used a 'simple' model due to the relatively low number of sample size in order to prevent over fitting. In the future if significantly more data is available it may be a good idea to use richer models such as neural networks or Bayesian networks that may improve our prediction models and outcomes. Last limitation has to do with the fact that the data was taken from maximal incremental CPET, which means that the subjects did not actually perform a real standard GXT. This might affect the accuracy of the results.

## Supporting information

**S1 Table. Features description.** Calculated features obtained from a GXT.
(PDF)

**S2 Table. Data of GXT features.**
(PDF)

## Acknowledgments

This study was carried out by Efrat Leopold under a PhD dissertation and by Prof Mickey Scheinowitz and Prof Tamir Tuller.

## Author Contributions

**Conceptualization:** Efrat Leopold, Tamir Tuller, Mickey Scheinowitz.

**Data curation:** Efrat Leopold, Mickey Scheinowitz.

**Formal analysis:** Efrat Leopold.

**Investigation:** Mickey Scheinowitz.

**Methodology:** Efrat Leopold, Tamir Tuller, Mickey Scheinowitz.

**Supervision:** Tamir Tuller, Mickey Scheinowitz.

**Validation:** Tamir Tuller, Mickey Scheinowitz.

**Visualization:** Tamir Tuller.

**Writing – original draft:** Efrat Leopold.

**Writing – review & editing:** Efrat Leopold, Tamir Tuller, Mickey Scheinowitz.

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
