## [Decision Letter · Decision Letter 0]

16 Aug 2022

PONE-D-22-16892A computational predictor of the anaerobic mechanical power outputs from a clinical exercise stress testPLOS ONE

Dear Dr. Scheinowitz,

Thank you for submitting your manuscript to PLOS ONE. After careful consideration, we feel that it has merit but does not fully meet PLOS ONE’s publication criteria as it currently stands. Therefore, we invite you to submit a revised version of the manuscript that addresses the points raised during the review process.

 Please submit your revised manuscript by Sep 30 2022 11:59PM. If you will need more time than this to complete your revisions, please reply to this message or contact the journal office at plosone@plos.org. Please include the following items when submitting your revised manuscript:A rebuttal letter that responds to each point raised by the academic editor and reviewer(s). You should upload this letter as a separate file labeled 'Response to Reviewers'.A marked-up copy of your manuscript that highlights changes made to the original version. You should upload this as a separate file labeled 'Revised Manuscript with Track Changes'.An unmarked version of your revised paper without tracked changes. You should upload this as a separate file labeled 'Manuscript'.

We look forward to receiving your revised manuscript.

Kind regards,

Emiliano Cè

Academic Editor

PLOS ONE

Journal Requirements:

2. Please clarify the relationship between the authors and the institution which approved the study and the institution where the research was carried out. We typically expect IRB approval from an institution with which at least one author is affiliated.

 "This study was partially supported by the Elizabeth and Nicholas Slezak Super Center for Cardiac Research and Medical Engineering, Tel Aviv University (MS)."  

    "This study was partially supported by the Elizabeth and Nicholas Slezak Super Center for Cardiac Research and Medical Engineering, Tel Aviv University (MS)"

 " This study was partially supported by the Elizabeth and Nicholas Slezak Super Center for Cardiac Research and Medical Engineering, Tel Aviv University (MS)."

6. Please include your tables as part of your main manuscript and remove the individual files. Please note that supplementary tables (should remain/ be uploaded) as separate "supporting information" files

Additional Editor Comments:

one expert in the field reviewed your manuscript detecting several major methodological issues you should consider during the revision process.

Reviewers' comments:

Reviewer's Responses to Questions

**Comments to the Author**

1. Is the manuscript technically sound, and do the data support the conclusions?

Reviewer #1: Yes

2. Has the statistical analysis been performed appropriately and rigorously? 

Reviewer #1: Yes

3. Have the authors made all data underlying the findings in their manuscript fully available?

Reviewer #1: Yes

4. Is the manuscript presented in an intelligible fashion and written in standard English?

Reviewer #1: Yes

5. Review Comments to the Author

Reviewer #1: A computational predictor of the anaerobic mechanical power outputs from a clinical exercise stress test

General

Order of authors is not consistent through submission.

A main issue is that you did not perform GXT but you used parameters from CPET test. CPET test require wearing a mask that may affect the exercise parameter, thus I am not convinced that it is meaningful using these adapted variables.

Introduction

Page 9, lines 4-6- The most common standard graded exercise test (GXT) include 5 incremental increase of intensity which terminate when a heart rate (HR) of 85% to 100% of 6 age-adjusted maximal HR is achieved [5].

The reference does not support the information in the sentence, please change reference.

Page 9, line 7- Numerous studies have previously shown….

Please provide references for this sentence.

Page 9, line 10- Since measuring phosphagen and glycogen contents is complicated, these metabolic systems are usually reflected via power output measurement using the WAnT.

Please provide references for this sentence.

Page 9, line 12- Therefore, since knowing the anaerobic components is important, performing an 13 anaerobic exercise test is required.

Why is knowing the anaerobic component is important? Please explain or provide reference. There is some explanations in line 19, but it should be clearer.

Page 9, line 21- This information is of 22 special interest for athletes.

Athlete’s population is not usually performing GXT, as you pointed yourself that this test is usually performed on non- health population.

Page 10, line 34- Anaerobic metabolism becomes dominate and allows the generation of additional mechanical power output.

References do not support the data in the session, please provide references that are more relevant.

Page 10, line 42- outputs…. Delete one

Page 11, line 63- References 17, 18 should be inserted after: “which is considered as the most popular anaerobic test”.

Page 12, line 93- Based on this, we have extracted 25 new features representing standard GXT as detailed in supplementary Table S1.

Please indicate the 25 features in the text.

Page 13, line 122- Please elaborate on the equation, for example, what Y stands for?

Page 13, line 122- “Prediction of MP using features taken from CPET produced a spearman correlation 177 coefficient of r = 0.91 which is lower than the MP predictions using features taken from 178 standard GXT”

Please explain, why is it lower in CPET?

Page 16, line 191- “This feature might represent the aerobic contribution during the WAnT”.

What feature did you mean? Also, reference 22 does not show anything about aerobic contribution….

Page 16, line 197- “Therefore, this feature can give an indication of lactate accumulation following glycogen metabolism, which is used during the WAnT”.

Again, what feature did you mean? What is the importance of this information?

6. PLOS authors have the option to publish the peer review history of their article (what does this mean?). If published, this will include your full peer review and any attached files.

Reviewer #1: No

---

## [Author Response · Author response to Decision Letter 0]

12 Dec 2022

Reviewer #1:

1. General - Order of authors is not consistent through submission.

We have changed it accordingly. 

2. A main issue is that you did not perform GXT but you used parameters from CPET test. CPET test require wearing a mask that may affect the exercise parameter, thus I am not convinced that it is meaningful using these adapted variables.

We agree with the reviewer and indeed wearing a mask during CPET may affect the results. However, since the developed prediction model included data from within the same individual who underwent both tests (aerobic and anerobic) we can assume that even if such an error exists, it would affect the entire population. 

Introduction

3. Page 9, lines 4-6- The most common standard graded exercise test (GXT) include 5 incremental increase of intensity which terminate when a heart rate (HR) of 85% to 100% of 6 age-adjusted maximal HR is achieved [5].

The reference does not support the information in the sentence, please change reference. 

We have revised the sentence accordingly and added another reference: 

“A standard graded exercise test (GXT) include incremental increase [5] which terminate at submaximal or maximal intensities based on age-adjusted maximal HR [6].”

[6] Noonan V, Dean E. Submaximal exercise testing: clinical application and interpretation. Physical therapy. 2000 Aug 1;80(8):782-807.

4. Page 9, line 7- Numerous studies have previously shown….

Please provide references for this sentence. 

We have added the following references – 

[1] Bogdanis GC, Nevill ME, Boobis LH, Lakomy HK. Contribution of phosphocreatine and aerobic metabolism to energy supply during repeated sprint exercise. J Appl Physiol. 1996 Mar 1;80(3):876-84.

[2] Bogdanis GC, Nevill ME, Lakomy HK, Boobis LH. Power output and muscle metabolism during and following recovery from 10 and 20 s of maximal sprint exercise in humans. Acta Physiol Scand. 1998 Jun;163(3):261-72.

[3] Green S. Measurement of anaerobic work capacities in humans. Sports Medicine. 1995 Jan;19(1):32-42.

5. Page 9, line 10- Since measuring phosphagen and glycogen contents is complicated, these metabolic systems are usually reflected via power output measurement using the WAnT.

Please provide references for this sentence. 

We have revised the sentence and added more references –

Since measuring phosphagen and glycogen contents is complicated and required muscles samples (1-2), these metabolic systems are usually reflected via power output measurement using the WAnT.

[1] Barnett C, Carey M, Proietto J, Cerin E, Febbraio MA, Jenkins D. Muscle metabolism during sprint exercise in man: influence of sprint training. J Sci Med Sport. 2004; 7: 314-22.

[2] Karlsson J, Saltin B. Lactate, ATP, and CP in working muscles during exhaustive exercise in man. J Appl Physiol. 1970; 29: 598-602.

6. Page 9, line 12- Therefore, since knowing the anaerobic components is important, performing an 13 anaerobic exercise test is required. 

Why is knowing the anaerobic component is important? Please explain or provide reference. There is some explanations in line 19, but it should be clearer. 

The information regarding the anaerobic components will provide a better understating about the subjects' physiological capabilities which is especially important for athletes, helping specify the aerobic and anaerobic contribution for exercise training recommendations using a single exercise stress test. Furthermore, assessments of the anaerobic performance can help athletes, team players and individual sports by providing the coach with valuable information about the athletes’ fitness status as well as allowing them to monitor improvement through training.

This was added to the discussion section. 

7. Page 9, line 21- This information is of 22 special interest for athletes. 

Athlete’s population is not usually performing GXT, as you pointed yourself that this test is usually performed on non- health population. 

In our previous manuscript [1] we developed a prediction algorithm for the anaerobic mechanical power outputs driven from CPET. In the current study we wanted to test the hypothesis whether GXT is sufficient to predict the anaerobic mechanical power outputs. The implications of the current study are relevant for both athletes and normal individuals.

[1] Leopold E, Navot-Mintzer D, Shargal E, Tsuk S, Tuller T, Scheinowitz M. Prediction of the Wingate anaerobic mechanical power outputs from a maximal incremental cardiopulmonary exercise stress test using machine-learning approach. PLoS one. 14(3):e0212199, 2019.

8. Page 10, line 34- Anaerobic metabolism becomes dominate and allows the generation of additional mechanical power output.

References do not support the data in the session, please provide references that are more relevant.

We have changed the reference to the following one:

Scott CB. Contribution of anaerobic energy expenditure to whole body thermogenesis. Nutrition & Metabolism. 2005 Dec;2(1):1-9.

9. Page 10, line 42- outputs…. Delete one

We have deleted one accordingly. 

10. Page 11, line 63- References 17, 18 should be inserted after: “which is considered as the most popular anaerobic test”. 

We have changed it accordingly. 

11. Page 12, line 93- Based on this, we have extracted 25 new features representing standard GXT as detailed in supplementary Table S1.

Please indicate the 25 features in the text.

We have added this information into the text.

12. Page 13, line 122- Please elaborate on the equation, for example, what Y stands for?

Yreal refers to the real value of the predicted feature – either PP or MP.

Yprediced refers to the predicted value that is calculated – either PP or MP.

We have incorporated those changes into the revised manuscript in the materials and methods section.

13. Page 13, line 122- “Prediction of MP using features taken from CPET produced a spearman correlation 177 coefficient of r = 0.91 which is lower than the MP predictions using features taken from 178 standard GXT”

Please explain, why is it lower in CPET? 

The prediction correlations are similar in both CPET and GXT (only slightly higher by 0.04%, in features taken from a standard EST). Since eventually the GXT features were extracted from the CPET data, the prediction correlation for the GXT was high. 

Page 16, line 191- “This feature might represent the aerobic contribution during the WAnT”.

What feature did you mean? Also, reference 22 does not show anything about aerobic contribution…. 

We meant the VO2max feature and changed that in the text.

We have also changed the reference to the following one: 

Smith JC, Hill DW. Contribution of energy systems during a Wingate power test. Br. J. Sports Med. 1991; 25: 196-9.

14. Page 16, line 197- “Therefore, this feature can give an indication of lactate accumulation following glycogen metabolism, which is used during the WAnT”.

Again, what feature did you mean? What is the importance of this information?

We have meant to the ventilatory anaerobic threshold feature and have changed it in the text accordingly. 

In this paragraph we attempted to explained the rationale behind the chosen aerobic features and the physiological reasons of why these features best predict the anaerobic mechanical power outputs. 

This is information is important in order to understand the physiological explanation behind of the chosen features.

---

## [Editor Report · Decision Letter 1]

19 Dec 2022

PONE-D-22-16892R1A computational predictor of the anaerobic mechanical power outputs from a clinical exercise stress testPLOS ONE

Dear Dr. Scheinowitz,

Thank you for submitting your manuscript to PLOS ONE. After careful consideration, we feel that it has merit but does not fully meet PLOS ONE’s publication criteria as it currently stands. Therefore, we invite you to submit a revised version of the manuscript that addresses the points raised during the review process.

ACADEMIC EDITOR:Dear Authors,your manuscript has been reviewed by one expert in the field that retrieved several major issues to be considered in the revision process. Please submit your revised manuscript by Feb 02 2023 11:59PM. If you will need more time than this to complete your revisions, please reply to this message or contact the journal office at plosone@plos.org. Please include the following items when submitting your revised manuscript:A rebuttal letter that responds to each point raised by the academic editor and reviewer(s). You should upload this letter as a separate file labeled 'Response to Reviewers'.A marked-up copy of your manuscript that highlights changes made to the original version. You should upload this as a separate file labeled 'Revised Manuscript with Track Changes'.An unmarked version of your revised paper without tracked changes. You should upload this as a separate file labeled 'Manuscript'.

We look forward to receiving your revised manuscript.

Kind regards,

Emiliano Cè

Academic Editor

PLOS ONE

---

## [Author Response · Author response to Decision Letter 1]

21 Jan 2023

Following the email we received, we included the three items in the revised version.

---

## [Editor Report · Decision Letter 2]

13 Mar 2023

A computational predictor of the anaerobic mechanical power outputs from a clinical exercise stress test

PONE-D-22-16892R2

Dear Dr. Scheinowitz,

We’re pleased to inform you that your manuscript has been judged scientifically suitable for publication and will be formally accepted for publication once it meets all outstanding technical requirements.

Kind regards,

Emiliano Cè

Academic Editor

PLOS ONE
---

## [Editor Report · Acceptance letter]

25 Apr 2023

PONE-D-22-16892R2 

A computational predictor of the anaerobic mechanical power outputs from a clinical exercise stress test 

Dear Dr. Scheinowitz:

I'm pleased to inform you that your manuscript has been deemed suitable for publication in PLOS ONE. Congratulations! Your manuscript is now with our production department. 

Kind regards, 

on behalf of

Professor Emiliano Cè 

Academic Editor

PLOS ONE